# Nuclear mRNA Export and Aging

**DOI:** 10.3390/ijms23105451

**Published:** 2022-05-13

**Authors:** Hyun-Sun Park, Jongbok Lee, Hyun-Shik Lee, Seong Hoon Ahn, Hong-Yeoul Ryu

**Affiliations:** 1Department of Biochemistry, Inje University College of Medicine, Busan 50834, Korea; hspark@inje.ac.kr; 2Department of Biological and Chemical Engineering, Hongik University, 2639, Sejong-ro, Jochiwon-eup, Sejong-si 30016, Korea; jlee0917@hongik.ac.kr; 3BK21 FOUR KNU Creative BioResearch Group, School of Life Sciences, College of National Sciences, Kyungpook National University, Daegu 41566, Korea; leeh@knu.ac.kr; 4Department of Molecular and Life Science, College of Science and Convergence Technology, ERICA Campus, Hanyang University, Ansan 15588, Korea

**Keywords:** mRNA export, TREX, TREX-2, NPC, lifespan, neurodegenerative diseases

## Abstract

The relationship between transcription and aging is one that has been studied intensively and experimentally with diverse attempts. However, the impact of the nuclear mRNA export on the aging process following its transcription is still poorly understood, although the nuclear events after transcription are coupled closely with the transcription pathway because the essential factors required for mRNA transport, namely TREX, TREX-2, and nuclear pore complex (NPC), physically and functionally interact with various transcription factors, including the activator/repressor and pre-mRNA processing factors. Dysregulation of the mediating factors for mRNA export from the nucleus generally leads to the aberrant accumulation of nuclear mRNA and further impairment in the vegetative growth and normal lifespan and the pathogenesis of neurodegenerative diseases. The optimal stoichiometry and density of NPC are destroyed during the process of cellular aging, and their damage triggers a defect of function in the nuclear permeability barrier. This review describes recent findings regarding the role of the nuclear mRNA export in cellular aging and age-related neurodegenerative disorders.

## 1. Introduction

Eukaryotic transcription is a complex stepwise process comprised of the transcription initiation, elongation, and termination and requires multiple factors, including transcription machinery (RNA polymerase and general transcription factors), transcription cofactors (coactivator or corepressor), and chromatin regulators [1]. During the process of transcription, the nascent pre-mRNA is processed by 5′-end capping, removal of intron via splicing, and 3′-end cleavage and polyadenylation. The mature mRNA is then exported from the nucleus to the cytoplasm to undergo protein translation, and aberrantly processed pre-mRNAs and mRNAs are eliminated via the RNA surveillance system. Despite the distinct factors that carry out each of the steps in the pathway of gene expression, each factor interacts both physically and functionally with other proteins in the different pathways, coupling among the gene expression machineries [2].

Aging is a process that is accompanied by the progressive impairment at the molecular, cellular, and organ levels, eventually leading to the decay of biological and physiological functions and the increased risk of diverse aging-related diseases such as cancer, cardiovascular, and neurodegenerative diseases [3,4]. The rate and progression of aging are influenced by highly complex and diverse genetic and environmental factors, and the transcription process is linked closely to aging and its related disorders [5].

The transcription of genes triggers an increased opportunity for damage or mutation to affect DNA because the transcription machinery-mediated unwinding of the DNA double helix leads to exposure of single strand DNA to mutagenic agents [6]. The transcription’s fidelity is remarkably impaired with aging, contributing to genotoxicity and proteotoxicity and the eventual reduction of cellular longevity [7]. The transcription errors are not always random or temporary and often mimic DNA mutations, frequently inducing genetic diseases. For example, the transcription errors in the genes encoding UBB and APP lead to translation of the toxic forms of ubiquitin-B and amyloid precursor proteins in patients with Alzheimer’s disease [8,9], and the 8-Oxoguanine-mediated transcription errors in the *RAS* gene can induce the oncogenic pathway in mammalian cells [10]. Furthermore, such errors occasionally cause proteins to be misfolded, which can escape recognition by protein quality control machinery and survive inside cells for extended periods of time [11]. In various species, tissues, and cell types, aging is associated with alterations in the expression of diverse genes involved in signaling pathways, genetics, translational mechanisms, and metabolism, and its maintenance is critical for normal functioning to continue [5]. Although transcription is an essential process for life and survival, transcription itself and its mis-regulation cause genome instability and premature aging.

Similarly, the correlation between transcription and aging has been studied intensively through various experiments. However, extraordinarily little is known about the effects exhibited on the aging process as a result of nuclear events after transcription takes place, and such research has not been focused up to date. Thus, this review provides an initial overview of the current studies and recent progress in elucidating the role of the nuclear RNA export in cellular aging and the pathogenesis of neurodegenerative disorders.

## 2. Nuclear mRNA Export Pathway

In eukaryotic cells, the nuclear export of mRNA transcripts requires multiple cellular events including transcription, maturation of pre-mRNA, and the assembly of mature mRNA with specific RNA binding proteins, an establishing messenger ribonucleoprotein (mRNP) complex, and the mRNA transport through the nuclear pore complexes (NPCs) into the cytoplasm [12].

The process of mRNA processing is cotranscriptionally coupled to the mRNA transport pathway. In particular, 3′-end mRNA processing is clearly linked with the mRNA export, and the involved factors are evolutionary conserved from yeast to humans [13,14] (Figure 1). In yeast, polyadenylation leads to the recruitment of the poly(A) binding protein Nab2 (ZC3H14 in human) and its binding partner Yra1 (ALY in human), interacting directly with the essential mRNA export receptor Mex67-Mtr2 (NXF1-NXT1 in human) [15,16,17,18]. In addition, the recruitment of Yra1 is dependent on the interaction with Pcf11, an essential component of cleavage and the polyadenylation factor IA, which then transfers Yra1 to the transcription/export (TREX) complex with the aid of the Sub2 helicase (UAP56 in human) [19,20,21,22]. The THO proteins (Tho2, Hpr1, Mft1, Thp2, and Tex1), a core member of the TREX complex, and the Sub2 helicase are required for efficient polyadenylation by the Pap1 Poly(A) polymerase, indicating coupling among polyadenylation, dissociation of the polyadenylation proteins, and the release of the mRNP from the transcription unit [23,24,25]. Recently, it was also reported that two distinct ALY-interacting factors, NXF1 and TREX, prefer selectively to export different transcript groups depending on exon architecture and G/C content in human cells [26]. Additionally, mammalian SR proteins known as an alternative pre-mRNA splicing factor promote NXF1 recruitment to mRNA, and this interaction suggests a link between alternative splicing and the mRNA export, thereby controlling the cytoplasmic abundance of transcripts with alternative 3′ ends [27].

The transcription and export complex-2 (TREX-2), composed of Sac3, Thp1, Cdc31, Sem1, and Sus1, physically and functionally interacts with both the Spt-Ada-Gcn5 acetyltransferase (SAGA) transcription coactivator complex and NPC to link the transcription, mRNA export, and targeting of active genes to NPC [28] (Figure 2A). The N-terminus in Sac3 acts as a scaffold for association with Thp1 and Sem1, creating an mRNA-binding module, and with the mRNA exporter Mex67-Mtr2, whereas its C-terminus binds to Sus1, Cdc31, and Nup1 nucleoporin, providing a docking platform at NPC [29,30,31,32,33]. TREX-2 shares one subunit Sus1 with the DUB module for deubiquitination of H2B in the SAGA complex, and Sus1 simultaneously associates with the promoter and coding regions of some SAGA-dependent genes, offering a functional link of the transcription activation to the mRNA export [34,35,36,37]. Specifically, the Sus1, Sac3, and Thp1 subunits facilitate the post-transcriptional anchoring of transcribed genes to NPC upon the activation of transcription [38,39,40]. Both Cdc31 and Sem1 also contribute synergistically to mediate the association of TREX-2 with NPC for promoting the mRNA export process [31,41]. In human TREX-2, a germinal center associated nuclear protein (GANP), known as Sac3 orthologue, is associated with the RNA polymerase II and Nxf1 (Mex67 in yeast), facilitating the movement of mRNP to NPC [42]. Inhibition of the processing of mRNA leads to the redistribution of GANP from NPC into nuclear foci, suggesting that TREX-2 mediates the transportation of the mRNP from active genes to NPC [43].

## 3. NPC and mRNA Export

The eukaryotic NPC is composed of about 30 nucleoporin proteins and has a radial symmetry of eightfold. Its structure is composed of three main parts, a central core spanning the nuclear envelope (NE) membrane, a nuclear basket, and long cytoplasmic filaments, and selectively allows most of the mRNPs to disperse in and out of the nucleus in order to maintain a barrier of nuclear permeability [44]. The transmembrane nucleoporins physically tether NPC to the NE membrane, while structural nucleoporins, embedded in the NPC, serve as a platform for the other nucleoporins and FG-nucleoporins containing phenylalanine–glycine (FG)-repeats, such as FG, FXFG, and GLFG [45,46]. The symmetrical FG-nucleoporins are located on the both sides of the NPC, while the asymmetrical FG-nucleoporins are observed exclusively on one side of the NPC [46].

The mRNP anchors to the NPC by interacting directly between the mRNA export factors and basket nucleoporins located in the nucleoplasmic region [12]. In yeast, the FG-repeats of Nup49, Nup57, Nup1, and Nup2 nucleoporins provide the first docking sites for mRNP to NPC via their interaction with Mex67 [47]. However, when the transport route of single native mRNA particles was monitored in insect and human cells, 60–75% of them are able to return to the interchromatin region after association with the basket [48,49]. Furthermore, when the export procedure was inhibited by the treatment of wheat germ agglutinin (WGA) in human cells, an accumulation of mRNPs at the nuclear periphery was found, suggesting that the interaction between the mRNP and NPC is independent of the export process [50]. Another plausible explanation is that such nucleoplasmic mRNP flux may function as a rate-limiting step at the NPC basket by spending for a long duration before reaching the mRNP to the NPC. When the imaging study revealed single native mRNA particles moving across the NE in insect cells, only 25% of the encounter particles with NE were successfully sent to the cytoplasm [48]. Additionally, monitoring the actual flow of the β-actin mRNA revealed that the rate-limiting steps for the nucleocytoplasmic transport of the mRNP are both the access and the release from the NPC [51]. Therefore, the quality control and surveillance mechanisms for mRNA are estimated to be important pathways for the rate-limiting step [12].

## 4. mRNA Export Factors and Aging

The connection between gene expression and aging is reflected in the diverse transcription factors that can operate as the key factors in regulating the various cellular processes [52,53,54,55,56,57,58,59,60]. Among such factors involved in the regulation of lifespan, the SAGA complex, a physical and functional partner of TREX-2, has multiple roles depending on its independent modules, HAT module (histone acetylation), DUB module (deubiquitination of H2B), TAF module (coactivator architecture), and SPT module (assembly of the preinitiation complex), in yeast aging pathway [61] (Figure 2B). The presence of a HAT inhibitor, inducing a low level of histone acetylation, leads to an extended replicative lifespan (RLS) which is completely abolished upon the loss of Gcn5, a catalytic subunit of the SAGA HAT module [62]. A RLS is significantly also extended in the presence of the heterozygous mutant *gcn5* or *ngg1*, a gene encoding a linking protein between Gcn5 and SAGA [62,63], whereas each loss of other components in the SAGA HAT module does not lead to an increase in the yeast lifespan [64,65]. A loss of Ubp8, Sgf73, or Sgf11 in the SAGA DUB module greatly extends a RLS in a Sir2-dependent mechanism for maintaining telomeric silencing and rDNA stability, the most representative pathway for controlling the lifespan of yeast [65], while both a RLS and the chronological lifespan (CLS) are mostly decreased in the cells lacking each component in the SAGA SPT module [64]. In addition, SAGA promotes anchoring the non-chromosomal DNA circles to the NPC and concomitantly leads to confinement of such circles in the mother nucleus, which is a characteristic feature of aged nucleus [66]. Although it is still unclear how a single complex has multiple functions that ensure a normal lifespan, SAGA is a good example of how aging is finely tuned by regulators in a complex network.

The THO complex is required for the environmental stress response and maintaining a normal fly lifespan. Mutations in the THO complex resulted in a shortened lifespan and strong sensitivity to certain environmental stressors. This is suppressed by the upregulation of c-Jun N-terminal kinase signaling which regulates stress tolerance and longevity [67]. Genome-wide transcriptomic analyses revealed that the gene expression of TREX and other factors that are required for trafficking nucleocytoplasma were globally downregulated in five distinct types of senescent cells, representing replicative senescence, tumor cell senescence, oncogene-induced senescence, stem cell senescence, and progeria and endothelial cell senescence. Such a similar enrichment pattern was observed in two large human tissue genomic databases: Genotype-Tissue Expression and The Cancer Genome Atlas [68]. Furthermore, the enrichment patterns of TREX and NPC-related factors were conversely upregulated during the process of tumorogenesis, suggesting that the failure of age-related changes in gene expression profile of TREX and related factors may lead to an increased risk for aging-related cancer [68].

A very recent study revealed that TREX-2 is also involved in the maintenance of a normal lifespan in yeast [69]. The loss of two major structural components of TREX-2, Thp1 and Sac3, and a linker protein Sus1 between the SAGA DUB module and TREX-2 impaired the normal lifespan and vegetative growth. In particular, TREX-2 regulates the RLS in a Sir2-independent manner, and the growth and lifespan defects by the loss of Sus1 were the fault of TREX-2 rather than the SAGA DUB. Moreover, the growth defect, shortened lifespan, and nuclear accumulation of poly(A)^+^ RNA in cells lacking Sus1 were rescued by an increased dosage of the mRNA export factors Mex67 and Dbp5, whose association with the nuclear rim was affected by Sus1, suggesting that boosting the mRNA export process restores the defect of mRNA transport and further damage in the growth and lifespan by lack of Sus1 (Figure 3). In short, an abnormal accumulation of nuclear RNA is a negative factor for ensuring a normal lifespan.

## 5. NPC and Aging

The age-dependent deterioration of nucleoporins accelerates the damages in the structure and function of the NPC, leading to the loss of the barrier of nuclear permeability and a leakage of cytoplasmic proteins into the nucleus [70]. In differentiated rat brain cells, nucleoporins are oxidized and long-lived without a turnover of the NPC via the degradation of old proteins and a new synthesis which results in definite harmful effects [70,71]. In yeast, the correlation between NPCs and the lifespan of cells was directly analyzed by a RLS measurement method [72]. The RLS was impaired by lacking the GLFG domain of Nup116, while such a shortened lifespan is rescued by the overexpression of Gsp1, the small GTPase that facilitates the karyopherin Kap121-mediated transport. However, the Nup100-mediated control of the tRNA life cycle potentially limits the yeast lifespan [72,73].

The optimal stoichiometry and density of NPC are disrupted during the process of aging [74]. The senescent human fibroblasts exhibit several characteristic features, such as hypo-responsiveness either to growth factors or to apoptotic signals that are induced by diverse stimuli [75,76,77,78] and a decreased cellular level of nucleocytoplasmic transport factors, Nup88, Nup107, Nup155, Nup50, karyopherin, Ran (Ras-related GTPase), and Ran-regulating factors, suggesting that senescence-associated hypo-responsiveness would be the result from a reduction in the nuclear translocation by the loss of the stoichiometry of nucleocytoplasmic transporters [79]. Such an alteration in the optimal level of the NPC is similarly observed in older yeast cells [80,81]. Although another senescent phenotype is the changed distribution and density of the NPC at NE, reflecting irregular nuclear organization and function [82], it is unclear how the density of the NPC increases its effects on the longevity and the process of aging.

## 6. MRNA Turnover and Aging

The mRNA surveillance process ensures that the properly processed transcripts are present within the cell and are coupled to the mRNA export pathway [83], and its defects often drive cellular senescence [84]. For instance, a decrease in the human RNA turnover rate via the declined activity of the RNA exosome or oxidative stress triggers cellular senescence [85]. The expression of the *PHO84* gene is repressed by the corresponding antisense RNA in chronologically aged yeast cells, and stabilization of the antisense RNA is facilitated by the Rrp6/exosome complex and histone acetylation [86]. In addition, unspliced or malformed transcripts are identified and degraded during the quality control step involved with certain nucleoporins, endonuclease Swt1, and protease Ulp1 upon the docking of the mRNP to the basket of NPC in yeast [87,88]. Therefore, the mRNA turnover mechanism inhibits nuclear accumulation and the abnormal export of misprocessed RNA species and is important in preventing pathophysiological cell senescence and cell death.

## 7. MRNA Export and Age-Related Neurodegenerative Disorders

A defective mRNA export is implicated in diverse neurodegenerative disorders [89,90]. The mislocalization of the THO complex subunit two (THOC2) to the cytoplasm was detected in HEK293T cells that were transfected with Htt96Q or TDP-43 associated with Huntington’s disease (HD) and amyotrophic lateral sclerosis (ALS), respectively [91]. The mutations of THOC4 act as a potential neurodegeneration suppressor in the fly ALS model [92], while a knockout of THOC5 in mouse dopaminergic neurons leads to a defect in the nuclear export of synaptic transcripts and degeneration of the neurons, leading to the death of the animal [93]. Matrin3, a protein associated with ALS, interacts physically with multiple TREX proteins, and its mutations cause the nuclear mRNA export defects of both the global mRNA and ALS-related transcripts in particular [94]. In addition, TDP-43 itself binds thousands of introns and 3′ UTRs of pre-mRNAs, and its mutations lead to abnormal localization of nucleoporins and the nuclear retention of poly(A)^+^ RNA [95,96]. Similar to TDP-43, FUS, whose mutations cause ALS, also associates with thousands of mRNAs, and it appears to promote mRNA export in neural dendrites [97,98,99].

The abnormal aggregation of two scaffold nucleoporins Nup205 and Nup107, mislocalization of Nup62 at NE, and mutations in the GLE1 gene encoding the nuclear mRNA export factor that physically interacts with NPC were found in patients with ALS [100,101,102]. A loss of Nup358 (also called E3 SUMO-protein ligase RanBP2) in murine motoneurons drives the ALS-like syndrome, suggesting that the irregular composition and distribution of nucleoporin might play an important role in ALS pathophysiology [103]. Similar to ALS, abnormal localization of Gle1, Nup62, and RanGAP1 (the binding partner of Nup358) in multiple models of HD and Nup62 in the hippocampus and neocortex of Alzheimer’s disease patients was previously reported [104,105,106]. Parkin, whose mutations are considered to be one of the most common causes of the familial Parkinson’s disease, selectively binds to Nup358 and promotes its degradation [107,108]. In addition, defects in the export of the mitochondrial mRNA through NE budding, a distinct pathway with the NPC-mediated nucleocytoplasmic transport, displayed progressive mitochondrial disruption, resulting in accelerated aging [109]. Taken together, these studies indicate that the disruption of the nucleocytoplasmic transport is a central feature of neurodegenerative diseases. Studying the function of the mRNA export in aging may provide clues for developing new therapies that can block neurodegeneration.

## 8. Conclusions

The link between gene transcription and aging has been well characterized in diverse studies. The gene expression profile is extremely changed in senescent cells, indicating the various biological events that occur during the process of aging [110], and the transcription itself is able to accelerate the rate of damage to DNA, leading to genomic instability and further premature aging [6]. The transcription error rates are increased with aging, inducing the aggregation of peptides that characterize age-associated disorders [11]. The nuclear events that occur after the process of transcription are also an important element related to cellular aging, and this process is closely coupled with the transcription progress. The essential factors required for the mRNA export, TREX, TREX-2, and NPC, are dynamically interacted with a number of transcription factors, including SAGA and pre-mRNA processing factors [14,19,20,22,24,28,42], and the mutation of such factors involved in the transportation of mRNA generally triggers the accumulation of the nuclear RNA by blocking a release of the mRNA into the cytoplasm and further shortened lifespan [15,17,18,19,20,29,30,31,32,33,36,47,69,72,73,93]. However, because the study of a defect in the mRNA export was focused on monitoring a single native mRNA molecule or poly(A)^+^ RNA, genome-wide analysis approaches may assist in uncovering whether nucleoplasmic trafficking of specific RNA transcript(s) affects cellular lifespan. In addition, the induction of the smooth mRNA transport by an increased dosage of Mex67 or Dbp5 rescues the decreased lifespan in *sus1Δ* cells, implying that the prevention of nuclear RNA accumulation plays an important role in cellular aging [69]. Therefore, a change in the localization, stoichiometry, and density of the mRNA export factors may potentially hold value as a new marker for the detection of cellular aging and the study of longevity.

The histones are subject to multiple PTMs, including acetylation, methylation, phosphorylation, ubiquitination, and sumoylation, and such patterns of PTM constitute codes that regulate elaborate chromatin-based processes [111,112,113,114,115]. Not only SAGA-mediated regulation of deubiquitination and acetylation on histones but also the diverse modifications on histones is able to influence the pathway of the mRNA export. For example, yeast Mog1, a Ran GTPase-binding protein required for the nuclear protein import, maintains normal levels of the H2B ubiquitination and H3K4 methylation, and the mRNA export defect in *mog1Δ* is aggravated by the additional loss of factors for H2B ubiquitylation [116]. Direct interaction between the Setd2 H3K36 methyltransferase and Spt6/Iws1 transcription elongation complex may facilitate kinetics of the mRNA transport in human cell lines [117]. Additionally, because evidence has provided insight into the connection between histone modifications and aging [118], histone modifications may have a potential role in linking the nuclear RNA export to the lifespan. However, except for the SAGA complex having the activity of histone deubiquitination and acetylation, there are no available reports concerning the effects of the histone modification-mediated control of the mRNA export on the aging pathway. Hence, the better characterization of how histone modifications modulate the mRNA export from the nucleus to the cytoplasm may be a promising avenue for future research exploring the prevention of premature aging and the development of a new therapy for neurodegenerative disorders.

## Figures and Tables

**Figure 1 ijms-23-05451-f001:**
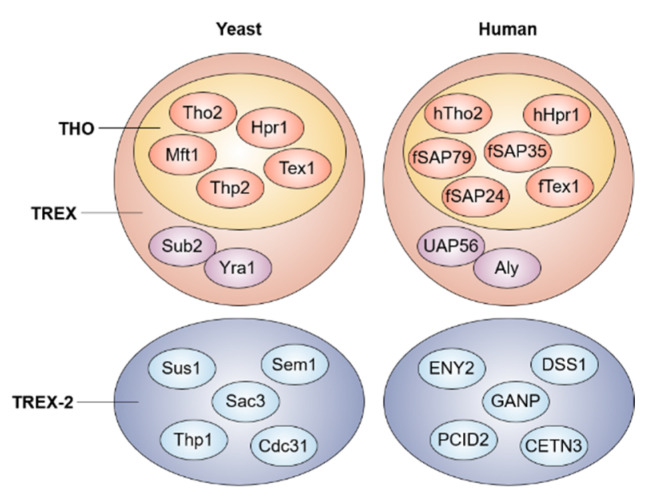
The conserved TREX and TREX-2 complexes. Both TREX and TREX-2 complexes are conserved between yeast and human. The TREX complex includes the multi-subunit THO complex and mRNA export proteins.

**Figure 2 ijms-23-05451-f002:**
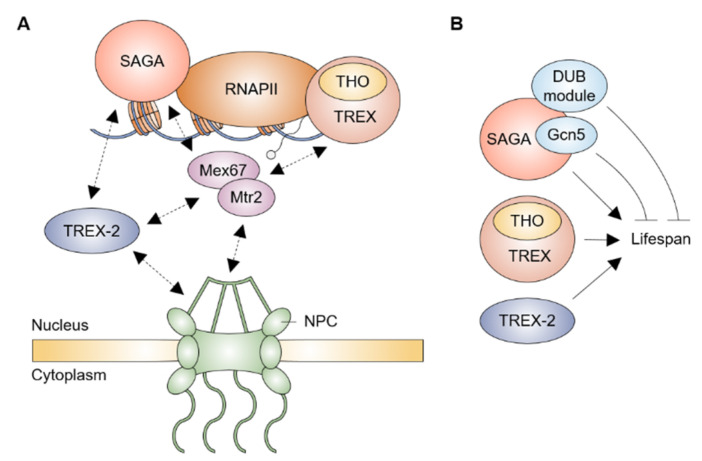
The effects of SAGA, TREX, and TREX-2 complexes on the yeast lifespan. (**A**) Model of SAGA, TREX, and TREX-2 complexes-mediated mRNA export pathway. At the stage of transcription initiation, SAGA is recruited to RNA polymerase II (RNAPII) machinery and mediates activation of transcription. TREX is co-transcriptionally recruited and associates with nascent transcripts. The mRNA export receptor Mex67-Mtr2 interacts with SAGA, TREX, and TREX-2 complexes and NPC, which facilitates passage of mature mRNP to cytoplasm. TREX-2 shares a component with SAGA and promotes anchoring of mRNP to NPC. Illustration reflects the relevant location of proteins but not precise physical association. (**B**) Many subunits in SAGA, TREX, and TREX-2 are required for blocking a shortened lifespan, whereas Gcn5 and DUB module (except for Sus1) in SAGA limit an abnormal extension of the lifespan.

**Figure 3 ijms-23-05451-f003:**
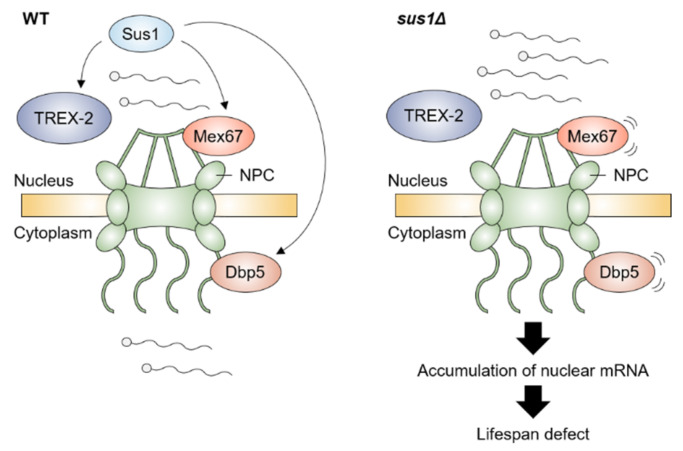
Sus1-mediated mRNA export pathway is required for maintaining a normal lifespan in yeast. In WT, Sus1, a component of TREX-2 complex, facilitates the proper association of Mex67 and Dbp5 with NPC, which requires efficient mRNA transport from nucleus to cytoplasm. In contrast, deletion of *SUS1* leads to mislocalization of Mex67 and Dbp5 and accumulation of nuclear mRNA, resulting in a further defect in the lifespan. Illustration reflects the relevant location of proteins but not their precise physical association.

## Data Availability

Not applicable.

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
