# Peer review of "Nuclear mRNA Export and Aging"

_ijms, 2022, doi:10.3390/ijms23105451_

Round 1
Reviewer 1 Report
The manuscript of Park et al. discusses the relationship between mRNA export defects and aging. In the first part the authors describe in detail the main molecular components involved in the transport of mRNA into the cytoplasm. These introductory data take up half of the publication.
The authors present data on the effect of mutations in genes encoding subunits of the SAGA, TREX, and TREX-2 complexes in yeast on aging. However, one of the major publications cited by the authors (and in which they are co-authors) has not yet passed the peer-review stage. The authors also cite data on the relationship between disruptions in NPC structure and function and cell aging, as well as on the association of the above factors with the development of neurodegenerative diseases. In their review, the authors focus on a very limited group of factors controlling post-transcriptional events in the nucleus.
It should be noted that by mRNA export, the authors apparently mean the whole set of processes from pre-mRNA synthesis to mature mRNA exit from the nuclear pore. A more accurate definition of export is necessary.
The factors mentioned in the publication are multifunctional and control a number of processes. SAGA modifies the chromatin template, TREX and TREX-2 control nuclear mRNA maturation, and NPC is required for transport of multiple molecules. This raises the question of the role of the mRNA transport process per se in aging phenotype development. The described mutations might disrupt nuclear mRNA maturation, which then manifests, in particular, in impaired transport of incorrectly processed RNA. Can the authors provide convincing evidence that disruption of the mRNA export step through the pore (while keeping other steps correct) has an independent effect on aging? What is the place of mRNA export disruption among other errors of gene expression stages in aging?
The data presented by the authors indicate that the process of transcription, maturation, export, and quality control of mRNA is disrupted with aging. The authors need to take a more comprehensive and in-depth view of the disruption of all of these processes during aging. Based on this broader view, a hypothesis about the significance and contribution of export disorders to aging can be formulated. The authors can also propose experimental ways to further address this issue.
Thus, the data presented in this publication are not yet sufficiently convincing to draw the conclusion taken as the title of the paper.
Undoubtedly this topic is novel and potentially of great interest as another molecular aspect of aging. The authors can provide a more in-depth and correct analysis of the issue.
Author Response
The manuscript of Park et al. discusses the relationship between mRNA export defects and aging. In the first part the authors describe in detail the main molecular components involved in the transport of mRNA into the cytoplasm. These introductory data take up half of the publication.
We thank the reviewer for the constructive critique and positive comments. mRNA export study is a complex and wide area of research, and it coupled delicately with transcription and RNA processing steps. Therefore, because we should explain accurately and concretely the process steps, the introduction part comprised a large proportion in this manuscript. In particular, the relationship between mRNA export and aging is a novel and creative topic, and, to our knowledge, ours is the first review paper on this topic. Given that most of the published results in this field have appeared only in the past few years, we believe this is an excellent time to summarize the state of the field.
The authors present data on the effect of mutations in genes encoding subunits of the SAGA, TREX, and TREX-2 complexes in yeast on aging. However, one of the major publications cited by the authors (and in which they are co-authors) has not yet passed the peer-review stage. The authors also cite data on the relationship between disruptions in NPC structure and function and cell aging, as well as on the association of the above factors with the development of neurodegenerative diseases. In their review, the authors focus on a very limited group of factors controlling post-transcriptional events in the nucleus.
We agree with referee’s comments, and some reports has now been newly added in the main text: “Recently, it was also reported that two distinct ALY-interacting factors, NXF1 and TREX, prefer selectively to export different transcript groups depending on exon architecture and G/C content in human cells.” (PMID 32504555) and “Additionally, mammalian SR proteins known as an alternative pre-mRNA splicing factor promote NXF1 recruitment to mRNA, and this interaction suggests a link between alternative splicing and the mRNA export, thereby controlling the cytoplasmic abundance of transcripts with alternative 3 ′ ends.” (PMID 26944680) in the part 2.
In addition, we have added newly a model of SAGA, TREX, and TREX-2 complexes-mediated mRNA export pathway showing their interaction in Figure 2 (Figure 2A), to help the readers for better understanding.
A reference at peer-review stage (Lim, S.; Liu, Y.; Rhie, B.-H.; Kim, C.; Ryu, H.-Y.; Ahn, S.H. Sus1 maintains a normal lifespan through regulation of TREX-2 complex-mediated mRNA export. bioRxiv 2022, 2022.2004.2007.487427, doi:10.1101/2022.04.07.487427.) will be acceptable for publication in Aging (SCIE, IF: 5.682), pending some minor revision based on comments of the reviewers, and it will be published without additional experiments soon. We would provide a letter of minor revision as an evidence.
This field is an initiation stage, and there are relatively few reports concerning this topic. Therefore, we anticipate this manuscript would arouse interest from lots of researchers for further reports and diverse experiments.
References: PMID 32504555 and PMID 26944680
It should be noted that by mRNA export, the authors apparently mean the whole set of processes from pre-mRNA synthesis to mature mRNA exit from the nuclear pore. A more accurate definition of export is necessary.
Thanks for referee’s comments. As I mentioned above, mRNA export pathway is a not simple but complex. The nuclear export of mRNA transcripts can be broken down into distinct stages: first, pre-mRNA is transcribed in the nucleus, where it is processed and packaged into messenger ribonucleoprotein (mRNP) complexes; second, the mRNPs are targeted to and translocate through nuclear pore complexes (NPCs) that are embedded in the nuclear envelope; and third, the mRNPs are directionally released into the cytoplasm for translation (PMID: 19494120).
In response to the referee’s comments, we have edited “In eukaryotic cells, the expression of genes requires multiple cellular events including transcription, maturation of pre-mRNA, and the assembly of mature mRNA with specific RNA binding proteins, an establishing messenger ribonucleoprotein (mRNP) complex, and the mRNA export through the nuclear pore complexes (NPCs) into the cytoplasm” to “In eukaryotic cells, the nuclear export of mRNA transcripts requires multiple cellular events including transcription, maturation of pre-mRNA, and the assembly of mature mRNA with specific RNA binding proteins, an establishing messenger ribonucleoprotein (mRNP) complex, and the mRNA transport through the nuclear pore complexes (NPCs) into the cytoplasm” in the revised manuscript to inform definition of mRNA export pathway more accurately.
Reference: PMID 19494120
The factors mentioned in the publication are multifunctional and control a number of processes. SAGA modifies the chromatin template, TREX and TREX-2 control nuclear mRNA maturation, and NPC is required for transport of multiple molecules. This raises the question of the role of the mRNA transport process per se in aging phenotype development. The described mutations might disrupt nuclear mRNA maturation, which then manifests, in particular, in impaired transport of incorrectly processed RNA. Can the authors provide convincing evidence that disruption of the mRNA export step through the pore (while keeping other steps correct) has an independent effect on aging? What is the place of mRNA export disruption among other errors of gene expression stages in aging?
This is a good point, which we had also considered. The relationship between transcription and aging is one that has been studied intensively and experimentally with diverse attempts. But the impact of the nuclear mRNA export on the aging process following its transcription is still poorly understood. However, dysregulation of the mRNA export factors including SAGA, TREX, TREX-2, and NPC generally leads to the aberrant accumulation of nuclear mRNA and further impairment in the vegetative growth and normal lifespan (PMID 32157780, PMID 7862114, PMID 33831401, PMID 25043177, PMID 19167330, PMID 27932586, PMID 32553207). Also, the optimal stoichiometry and density of NPC are destroyed during the process of cellular aging, and their damage triggers a defect of function in the nuclear permeability barrier (PMID 12730243, PMID 10781609, PMID 11238892, PMID 19903462, PMID 26422514, PMID 31157618, PMID 17074834).
In particular, rescue of abnormal accumulation of mRNA in cells lacking Sus1 leads to restore the impaired replicative lifespan, suggesting that boosting the mRNA export process restores the defect of mRNA transport and further damage in the growth and lifespan by lacking of Sus1 (Lim, et al. bioRxiv 2022). This is a direct evidence showing mRNA export pathway is important for ensuring a normal lifespan. Furthermore, the mRNA turnover mechanism inhibits nuclear accumulation and the abnormal export of misprocessed RNA species and is important in preventing pathophysiological cell senescence and cell death (PMID 33446491, PMID 18022365). Therefore, the mRNA export defect is distinguished with transcription error in aging, although we can not exclude aging-related transcripts are specifically affected by a defect in mRNA export.
References: PMID 32157780, PMID 7862114, PMID 33831401, PMID 25043177, PMID 19167330, PMID 27932586, PMID 32553207, PMID 12730243, PMID 10781609, PMID 11238892, PMID 19903462, PMID 26422514, PMID 31157618, PMID 17074834, PMID 33446491, PMID 18022365, Lim, et al. bioRxiv 2022
The data presented by the authors indicate that the process of transcription, maturation, export, and quality control of mRNA is disrupted with aging. The authors need to take a more comprehensive and in-depth view of the disruption of all of these processes during aging. Based on this broader view, a hypothesis about the significance and contribution of export disorders to aging can be formulated. The authors can also propose experimental ways to further address this issue.
This is also a good point. In response to the referee’s comment, we have added other evidences, “In addition, TDP-43 itself binds thousands of introns and 3’UTRs of pre-mRNAs, and its mutations lead to abnormal localization of nucleoporins and the nuclear retention of poly(A)+ RNA (PMID 21358640, PMID 29311743).” and “Similar to TDP-43, FUS, whose mutations cause ALS, also associates with thousands of mRNAs, and it appears to promote mRNA export in neural dendrites (PMID 27378374, PMID 16317045, PMID 15797031).”, to part 7 to support our message about age-related neurodegenerative disorders.
In addition, to suggest experimental ways to further address this issue, we described a potential role of histone modifications in linking the nuclear RNA export to the lifespan and added “However, because the study of a defect in the mRNA export was focused to monitor single native mRNA molecule or poly(A)+ RNA, genome-wide analysis approaches may assist to uncover whether nucleoplasmic trafficking of specific RNA transcript(s) affects cellular lifespan.” in part 8.
References: PMID 21358640, PMID 29311743, PMID 27378374, PMID 16317045, PMID 15797031
Thus, the data presented in this publication are not yet sufficiently convincing to draw the conclusion taken as the title of the paper.
Thanks for the advice. In response the referee’s comment, we revised the title of the paper from “nuclear mRNA export ensures a normal lifespan” to “nuclear mRNA export and aging”.
Undoubtedly this topic is novel and potentially of great interest as another molecular aspect of aging. The authors can provide a more in-depth and correct analysis of the issue.
We thank the reviewer for the positive evaluation and the constructive comments to extend several points in the paper. The objective of this submitted article was to present recently published studies dissecting the relationship between mRNA export pathway and aging and arouse interest from the researchers for in-depth understanding aging process. Therefore, we think it is timely to issue this topic for encouraging this research field.

Reviewer 2 Report
The review touches upon an interesting connection between mRNA export and aging. I believe this review is timely. Following are some suggestions:
* For part 2. Nuclear mRNA export pathway also see PMID 32504555
* Figures are not a strong side of this manuscript, they don’t add too much information, particularly figure 2. How about a summary figure regarding changes in the like in PMID 32553207.
* Figure 3 – shouldn’t Dbp5 be closer to the pore and not on the fibrils? What does Mex67 signify in the figure – isn’t it bound to mRNA? Or are you suggesting it’s nucleoporin-like behaviors as in PMID 31753862 and 31375530
* Line 20 – last “the” in sentence is not necessary
* Line 22 – “blockage of mRNA that is released into the cytoplasm” – that cannot be released
* Line 58 – “in order to avoid” – is this intentional or just a by product
* Line 91 – SRSF proteins are also important for mRNA export (PMID 26944680)
* Line 113 – “into the nuclear foci” – I think the foci have not been mentioned so “the” should be removed
* Line 127 – “The mRNP anchors to NPC” – to the NPC
* Line 130 – Isn’t this related to one specific type of RNA (vs a general conclusion)?
* Line 132 – “Although the export procedure was” – Although when the…
* Line 138 – “of the imaging of insects” – insect cells
* Line 142 – “accessed and released from the NPC” – the access and the release from
* Line 164 – “causing the organization of aged nuclei” – causing is not the right word
* Line 195 needs rephrasing. Also in some cases mRNAs are retained in the nucleus (PMID 26711333)
* Line 207 – “In the differentiated rat” – without “the”
* Line 250 – “The Matrin3,” – without “the”
* Line 264 – “in the mitochondrial export of the mRNA through NE budding” – what does this mean?
* Line 276 – “The nuclear events that occurs after the process of transcription is also an important” – occur (without s) and are also an important
Author Response
The review touches upon an interesting connection between mRNA export and aging. I believe this review is timely. Following are some suggestions:
We thank the referee for the positive evaluation and the constructive comments to extend some points in the paper.
* For part 2. Nuclear mRNA export pathway also see PMID 32504555
This is a good suggestion, and this issue has now been added in the revised manuscript: “Recently, it was also reported that two distinct ALY-interacting factors, NXF1 and TREX, prefer selectively to export different transcript groups depending on exon architecture and G/C content in human cells.”
* Figures are not a strong side of this manuscript, they don’t add too much information, particularly figure 2. How about a summary figure regarding changes in the like in PMID 32553207.
Thanks for the advice, and we have added newly a model of SAGA, TREX, and TREX-2 complexes-mediated mRNA export pathway showing their interaction in Figure 2 (Figure 2A). The previous Figure 2 was changed to Figure 2B, simplifying the complex effects of each complexes on the yeast lifespan, and this Figure may allow the readers to understand easily their roles.
* Figure 3 – shouldn’t Dbp5 be closer to the pore and not on the fibrils? What does Mex67 signify in the figure – isn’t it bound to mRNA? Or are you suggesting it’s nucleoporin-like behaviors as in PMID 31753862 and 31375530
This is a good point, which we had also considered. During mRNA export process, export factors including Dbp5 and Mex67 move dynamically, but not statically, and this localization change is not simple to explain. However, we didn’t intend to show exactly location or physical interaction of each factors in Figure 3. Instead, we highlighted the effect of Sus1 on the normal location, thereby leading to accumulation of nuclear mRNA and lifespan defect. Therefore, to focus to describe the correlation between Sus1 and TREX-2/Dbp5/Mex67 in our manuscript and avoid confusion of readers, we have added “Illustration reflects the relevant location of proteins but not precise physical association.” to Figure 3 legend in the revised manuscript.
* Line 20 – last “the” in sentence is not necessary
We agree with referee’s suggestion, and we have edited it in the revised manuscript.
* Line 22 – “blockage of mRNA that is released into the cytoplasm” – that cannot be released
In response to the referee’s comment, we have removed “via a blockage of mRNA that is released into the cytoplasm” for better understanding of our message.
* Line 58 – “in order to avoid” – is this intentional or just a by product
Thanks for catching this! we have edited “in order to avoid” to “which can escape recognition by” in the revised manuscript to describe the previous report more accurately.
* Line 91 – SRSF proteins are also important for mRNA export (PMID 26944680)
This is a good point, and this issue has now been added in the revised manuscript: “Additionally, mammalian SR proteins known as an alternative pre-mRNA splicing factor promote NXF1 recruitment to mRNA, and this interaction suggests a link between alternative splicing and the mRNA export, thereby controlling the cytoplasmic abundance of transcripts with alternative 3 ′ ends.”
* Line 113 – “into the nuclear foci” – I think the foci have not been mentioned so “the” should be removed
We agree with referee’s suggestion, and we have removed it in the revised manuscript.
* Line 127 – “The mRNP anchors to NPC” – to the NPC
We agree with this suggestion, and we have changed it in the revised manuscript.
* Line 130 – Isn’t this related to one specific type of RNA (vs a general conclusion)?
This is a good point. I have changed “However, 60%–75% of the mRNPs are able to return to the interchromatin region after association with the basket in both insect and human cells” to “However, when the transport route of single native mRNA particles was monitored in insect and human cells, 60%–75% of them are able to return to the interchromatin region after association with the basket.” in the revised manuscript to describe the previous reports more accurately.
* Line 132 – “Although the export procedure was” – Although when the…
Thank you for this suggestion. We have changed “Although the export procedure” to “Also, when the export procedure” in the revised manuscript.
* Line 138 – “of the imaging of insects” – insect cells
Thank you for referee’s suggestion!! We have edited “When a study of the imaging of insects revealed single native mRNA particles moving across the NE” to “When the imaging study revealed single native mRNA particles moving across the NE in insect cells” in the revised manuscript.
* Line 142 – “accessed and released from the NPC” – the access and the release from
We agree with referee’s suggestion, and we have edited it in the revised manuscript.
* Line 164 – “causing the organization of aged nuclei” – causing is not the right word
We agree with referee’s suggestion, and we have edited “In addition, SAGA promotes anchoring the non-chromosomal DNA circles to NPC, causing the organization of aged nuclei” to “In addition, SAGA promotes anchoring the non-chromosomal DNA circles to the NPC and concomitantly leads to confinement of such circles in the mother nucleus, which is a characteristic feature of aged nucleus” in the revised manuscript to explain the correlation between SAGA and aged nucleus.
* Line 195 needs rephrasing. Also in some cases mRNAs are retained in the nucleus (PMID 26711333)
In response to this comment, we have edited “In short, the successful mRNA transport events without an abnormal accumulation of nuclear RNA is essential to ensure a normal lifespan.” To “In short, an abnormal accumulation of nuclear RNA is a negative factor for ensuring a normal lifespan.” in the revised manuscript to understand our message more easily.
* Line 207 – “In the differentiated rat” – without “the”
Thanks for catching this, and we have removed it in the revised manuscript.
* Line 250 – “The Matrin3,” – without “the”
We thank the referee to find this. We have edited it in the revised manuscript.
* Line 264 – “in the mitochondrial export of the mRNA through NE budding” – what does this mean?
In response to the referee’s comment, we have edited “in the mitochondrial export of the mRNA through NE budding” to “In the export of the mitochondrial mRNA through NE budding, a distinct pathway with the NPC-mediated nucleocytoplasmic transport,” in the revised manuscript to describe the previous report more accurately. We intended to introduce that alternative pathway for nucleocytoplasmic communication distinct from the movement of material through the nuclear pore complex is also linked with aging.
* Line 276 – “The nuclear events that occurs after the process of transcription is also an important” – occur (without s) and are also an important
We also thank to find these errors, and these were revised correctly.
Round 2
Reviewer 1 Report
The authors provided changes to the text as suggested by the comments.
Overall, the publication is an initial formulation of the hypothesis on the contribution of mRNA export to the aging process. To date, very little data have been obtained on this phenomenon. However, this hypothesis is undoubtedly of interest and promising for further discussion. This paper may be recommended for publication as a pioneer in the field.